# Thermal Stability of Dispersions of Amino-Functionalized Silica in Glycol and in 50–50 Aqueous Glycol

**DOI:** 10.3390/molecules29112686

**Published:** 2024-06-06

**Authors:** Marta Kalbarczyk, Sebastian Skupiński, Marek Kosmulski

**Affiliations:** Laboratory of Electrochemistry, Lublin University of Technology, Nadbystrzycka 38, 20-618 Lublin, Poland; m.kalbarczyk@pollub.pl (M.K.); s.skupinski@pollub.pl (S.S.)

**Keywords:** heat-transfer fluid, particle size, zeta potential, ethylene glycol

## Abstract

Dispersions of amino-functionalized silica in ethylene glycol (EG) and in aqueous glycol show excellent stability at room temperature. Stability at elevated temperatures would be much desired with respect to their potential application as heat-transfer fluids. Amino-functionalized silica was dispersed in EG and in 50–50 aqueous EG by mass. HCl and acetic acid were added to enhance the positive ζ potential. The dispersions were stored at 40, 60, 80, and 100 °C for up to 28 days, and ζ potential and apparent particle radius were studied as a function of elapsed time. The particles showed a positive ζ potential in excess of 40 mV (Smoluchowski), which remained unchanged for 28 days. Such a high absolute value of ζ potential is sufficient to stabilize the dispersion against flocculation and sedimentation. The apparent particle radius in acidified dispersions was about 70 nm, and it was stable for 28 days. The particles were larger in pH-neutral dispersions. The apparent particle radius was about 80 nm in fresh dispersions and it increased on long storage at 80 and 100 °C.

## 1. Introduction

Ethylene glycol (EG) and its mixtures with water have lower freezing temperatures and higher boiling temperatures than water, thus they have wider temperature ranges of liquid state than water and most other molecular liquids. Moreover, they are non-flammable, moderately reactive, moderately toxic, and inexpensive. This makes them good candidates for heat-transfer fluids, and, indeed, they are widely applied as coolants in car radiators. Propylene glycol (PG), glycerol, and their mixtures with water have similar properties and they may be alternatives for EG as components of heat-transfer fluids. High viscosity and low thermal conductivity (as compared, for example, with pure water) are substantial disadvantages of the above liquids in context of their possible applications as heat-transfer fluids. The thermal conductivity can be improved by addition of nanoparticles to heat-transfer fluids. Such nanofluids are also more viscous than corresponding fluids without nanoparticles, so the advantage of higher thermal conductivity is achieved at the expense of an adverse effect of higher viscosity (cf. Table 1 for specific references). Nanofluids have been extensively studied. Many studies have been devoted to nanofluids based on solvents other than glycols or aqueous glycols (including water), but the present study is focused on nanofluids based on EG.

The physical properties of EG and of its mixtures with water, also at elevated temperatures, are well-known. The physical properties of nanofluids relevant to their potential application as heat-transfer fluids, including, but not limited to viscosity and thermal conductivity, also at elevated temperatures, can be found in the recent scientific literature. A few examples of such studies are summarized in Table 1.

Moreover, thermal conductivity was studied as a function of temperature in the following systems: graphene-40% EG nanofluids stabilized with ionic surfactants at 30–60 °C [12], Ag-coated ZnO-EG nanofluids (no surfactant added) at 25–55 °C [13], functionalized graphene-50% EG nanofluids with or without addition of PVP at 30–70 °C [14], silica-commercial coolant based on 50% EG nanofluids at 25–65 °C [15], commercial amine-functionalized graphene multiwall CNTs—a commercial coolant based on aqueous propylene glycol nanofluid at 30–50 °C [16], and graphene oxide–magnetite–titania–EG nanofluids (no surfactant added) at 25–60 °C [17]. Several examples of studies of thermal conductivity of EG- and propylene glycol-based nanofluids as a function of temperature from the literature are collected in [9].

Zeta potential as a function of temperature was studied for functionalized graphene-50% EG nanofluids with or without the addition of PVP at 20–65 °C [14]. Two recent reviews [18,19] have reported on the recent studies of heat-transfer fluids, including but not limited to heat-transfer fluids based on EG.

The above measurements were carried out with fresh dispersions. This is a generally accepted truth, that physical properties of solvents relevant to their potential application as heat-transfer fluids, e.g., viscosity and thermal conductivity, are stable in time, and even long storage at elevated temperatures will not affect these properties. In contrast, the physical properties of dispersions including heat-transfer nanofluids are less stable in time. Namely, aggregation and dissolution–recrystallization of colloidal particles and degradation of surfactants used to stabilize the dispersions lead to variation in overall physical properties of dispersions. For example, the auto-catalytic hydrolysis of SDS, which is a popular ionic surfactant is well-known. The changes in physical properties of heat-transfer nanofluids on aging are enhanced at elevated temperatures. Therefore, the physical properties measured in fresh dispersions are not necessarily relevant to aged dispersions, and measurements of such properties in aged dispersions are much desired. The literature reports on physical properties of aged glycol-based nanofluids are rare.

Stability of dispersions against coagulation and sedimentation is a crucial problem in heat-transfer nanofluids. Direct assessment of such a stability can be performed by measurements of turbidity, of absorption and scattering of radiation, and by gravimetric determination of the mass of the sediment, but such studies are tedious. Therefore, measurements of the ζ potential and of the size of particles in dispersion are often performed to estimate the stability [20]. The relationship between the stability of dispersions against coagulation and sedimentation and the ζ potential is complicated, but as a rule of the thumb, one can assume that a high absolute value of the ζ potential (>50 mV) will ensure the stability, and with a low absolute value of the ζ potential (<30 mV), the particles will quickly settle down. Numerous studies of the ζ potential in EG-based nanofluids have been published, but most studies were performed in fresh dispersions. Several examples of such studies are summarized in ref. [21].

Very few studies of the ζ potential in aged EG-based nanofluids have been published. Zeta potential as a function of elapsed time (at room temperature) was studied for MgO-50% EG nanofluids stabilized with various surfactants [7], for multiwall CNTs-80–100% EG nanofluids (no surfactant added) [8], for multiwall CNTs-propylene glycol nanofluids stabilized with various surfactants [9], and for amino-functionalized silica-aqueous EG (different EG concentrations) nanofluids [22]. The above studies were performed at room temperature, and the present authors are not aware of any publications reporting the effect of aging at elevated temperatures on the ζ potential of particles in glycol-based nanofluids. Aging at elevated temperatures is relevant to heat-transfer fluids, because the heat transfer involves a temperature gradient by definition. Usually, the physico-chemical processes, which may be responsible for alteration of the properties of dispersions, are faster at high temperatures, so the present study focused on the temperatures > 25 °C, and lower temperatures were not studied here.

In most studies of the ζ potential in EG-based nanofluids (cf. [21] for references), the solid particles were stabilized with ionic surfactants. Adsorption of the ionic surfactants at the solid–liquid interface produces high positive or high negative ζ potentials (>40 mV in absolute value), which are required for a high stability. Amino functionalization is an alternative method to produce highly charged particles, but this method has been seldom used in EG-based heat-transfer nanofluids. Commercial amino-functionalized graphene was used as a component of heat-transfer fluids [16].

Amino-functionalized silica is well-known. Unlike original silica, which is negatively charged at pH > 4, amino-functionalized silica is positively charged at neutral pH [22,23,24,25,26,27]. The positive ζ potential of amino-functionalized silica can be enhanced by acidification of dispersions. Our previous work [22] showed that amino-functionalized silica is also positively charged in acidified EG–water mixtures (10–100% EG). Moreover, the positive ζ potential of the particles was maintained for at least one month. The dispersions were aged at room temperature only. Development of the ζ potential on aging at elevated temperatures in EG-based nanofluids is presented in this study. To our best knowledge, this is the first study of the effect of aging at elevated temperature on the physical properties of EG-based nanofluids.

The term “thermal stability” in this paper refers to an operation range of nanofluids; that is, to the temperatures below their boiling points. This semantic detail has to be explained, because several authors have used the term “thermal stability” for thermogravimetric analysis of nanofluids based on aqueous EG up to several hundred °C [28]. Obviously, the solvent evaporated below 200 °C.

## 2. Results and Discussion

The BET (Brunauer–Emmett–Teller) specific surface area of functionalized silicas was 169 m^2^/g. This is ½ of the specific surface area of the original silica measured in our laboratory (319 m^2^/g), while the manufacturer reports 380 m^2^/g. The pH-dependent surface-charging of functionalized silica in water is illustrated in Figure 1.

The isoelectric point IEP of amino-functionalized silica in fresh dispersions was at pH 10, which is higher than most IEPs of amino-functionalized silicas reported in the literature [22,23,24,25,26,27]. The electrokinetic curves reported in Figure 1 for different ionic strengths (10^−3^–10^−1^ M NaCl) are very similar. Actually, the ζ potential was higher at low ionic strengths (here, 10^−3^ M), according to a common rule [29]. Namely, Smoluchowski equation used to convert the electrophoretic mobility into the ζ potential in this study underestimated the ζ potential, especially at low ionic strengths (especially in EG and aqueous EG without deliberate addition of electrolyte). With moderate values of *κa* (1–100), the ζ potential can be calculated for uniform spherical particles, but for polydispersed and irregularly shaped particles, a suitable equation is not available. The actual ζ potential in 10^−3^ M NaCl at pH 3–4 was probably about 50 mV. The IEP shifted to pH 9 in aged dispersions. A similar shift is reported in our previous study [22], while most other electrokinetic studies of amino-functionalized silicas are limited to fresh dispersions.

The apparent particle radii in dispersions of amino-functionalized silica are reported in Figure 2.

The instrument software calculated the particle size using a certain model, and the values obtained for polydispersed and irregularly shaped particles were not the actual radii, but we believe that the variations in apparent particle size properly reflect the degree of particle aggregation. The maximum in apparent particle size at pH 10 roughly matches the IEP (Figure 1), which confirms the correlation observed in other dispersions [29]. The particle size in 0.1 M NaCl at pH 3–7 was systematically higher than at lower ionic strengths. This result indicates aggregation of primary particles at high ionic strengths, even far from the IEP, which is again in line with the trend observed in other dispersions [29]. The lowest apparent particle radius of about 70 nm observed at low ionic strengths (here: 10^−3^ M) and far from the IEP represents primary particles, and apparent particle radii of several hundred nm represent aggregates.

Evolution of pH, the ζ potential, and particle size in three dispersions of amino-functionalized silica are presented in Figure 3, Figure 4 and Figure 5. The pH (Figure 3) was very stable in acidified dispersions. In the pH-neutral dispersion, the pH slightly increased within the first few days, probably due to the release of ammonia as a result of hydrolysis of amino groups.

Figure 4 shows that high positive ζ potential in acidified dispersions was maintained for at least 1 month. Interestingly enough, the ζ potential in dispersions acidified with acetic acid was higher than that in dispersions acidified with HCl in spite of higher pH in dispersions acidified with acetic acid (Figure 3). This effect can be explained by a higher ionic strength in HCl-acidified solution.

Fresh pH-neutral dispersion showed a positive ζ potential of 35 mV, but the positive ζ potential was depressed on aging, and after 1 month the sign of ζ potential was reversed to negative. Figure 4 shows that the ζ potential measured in fresh dispersion is not necessarily valid for aged dispersions. We emphasize again that the values shown in Figure 4 were calculated by means of Smoluchowski equation and they are underestimated.

Figure 5 shows that fresh and aged acidified dispersions of amino-functionalized silica contained primary particles with radii of about 70 nm. In contrast, the pH-neutral dispersion showed a high degree of aggregation. These results are in line with Figure 4 and with the aforementioned rule of thumb. With high absolute values of ζ potential, the dispersion consisted of primary particles, and with absolute values of ζ potential < 20 mV the particles were aggregated. Figure 5 suggests that in spite of a high IEP of amino-functionalized silica (Figure 1), its pH-neutral aqueous dispersions are not suitable for heat-transfer nanofluids due to particle aggregation, but their performance can be improved by acidification.

High positive ζ potential and low degree of aggregation in aged acidified aqueous dispersions of amino-functionalized silica (Figure 4 and Figure 5) does not imply similar behavior in EG. Therefore, analogous experiments were carried out in aqueous glycol (0–100% EG), except the pH was not adjusted or measured. The acidity of dispersions in glycol was characterized by the analytical concentration of acid. One series of pH-neutral (no acid added) and two series of acidic dispersions (50 μL of 36% HCl or 500 μL of glacial acetic acid per 100 mL of solvent) were examined for ζ potential and size of particles of amino-functionalized silica. The ζ potentials of amino-functionalized silica in pH-neutral dispersions in aqueous glycol are shown in Figure 6. Only selected aging times are shown in Figure 6 (and in the next figures) to avoid overcrowding of the symbols, and the results obtained at other aging times were very similar as those shown in the figures. The positive ζ potential observed in all fresh dispersions (0–100% EG) dropped within the first few hours, and further aging had a rather insignificant effect on the ζ potential. The positive ζ potential in aged dispersions linearly increased with glycol concentrations. In aged dispersions in 0–20% EG, the electrokinetic behavior of amino-functionalized silica was similar to aged pH-neutral dispersions in water (Figure 4); that is, the ζ potential was too low to protect the particles from coagulation and sedimentation. However, in 40–100% EG, the ζ potential was sufficient to stabilize aged dispersions against coagulation and sedimentation.

The ζ potentials presented in the Figures were calculated from the Smoluchowski equation, but in pH-neutral dispersions in aqueous EG, *κa* is <<100, and the Huckel equation is more suitable than the Smoluchowski equation. The Huckel equation produces the ζ potentials higher than Smoluchowski by a factor of 3/2. In other words, the values of ζ potentials presented in Figure 6 should be multiplied by about 3/2 to obtain the actual ζ potentials.

In contrast with Figure 6, the ζ potentials in acidified dispersions presented in Figure 7 and Figure 8 are rather insensitive to aging, irrespective of EG concentration. The variations in the ζ potentials in 80–100% EG in acidified dispersions represent a random scatter of results, rather than systematic changes. The 80–100% EG showed very high viscosity and low dielectric constant [22]; thus, a small variation in the electrophoretic mobility caused a substantial variation in the ζ potential. Unlike with aged pH-neutral dispersions (Figure 6), in which the ζ potential linearly increased with the EG concentration, the ζ potential in aged, acidified dispersions (Figure 7 and Figure 8) was rather insensitive to the EG concentration, and it was about 40 mV (Smoluchowski) in dispersions acidified with HCl and about 50 mV (Smoluchowski) in dispersions acidified with acetic acid. These values correspond to 60 and 75 mV, respectively, according to Huckel’s equation. Again, the results obtained in aqueous EG (Figure 7 and Figure 8) are similar to the results obtained in purely aqueous systems (Figure 4). The high absolute value of ζ potential in aged dispersions of amino-functionalized silica reported in Figure 7 and Figure 8 are in line with the low apparent particle radii, which ranged from 65 to 120 nm, and which were equal or slightly higher than the apparent radii of primary particles in water (Figure 2 and Figure 5).

The results reported in Figure 6, Figure 7 and Figure 8 are similar to analogous results reported in our previous study [22] for another specimen of amino-functionalized silica (obtained according to another recipe). It is likely that other amino-functionalized silicas will behave alike. The ζ potentials presented in Figure 6, Figure 7 and Figure 8 and apparent particle radii suggest that pH-neutral dispersions of amino-functionalized silicas in 40–100% EG and acidified dispersions of amino-functionalized silicas in 0–100% EG are suitable for heat-transfer fluids due to their stability against coagulation and sedimentation. The dispersions aged at room temperature (Figure 6, Figure 7 and Figure 8) were uniformly turbid and colorless over the entire experiment duration.

The stable behavior of dispersions at room temperature does not imply their stability at elevated temperatures. Therefore, we selected five systems for further studies at elevated temperatures. These systems included dispersions of amino-functionalized silica in the following solvents (45 mg of particles per 150 g of solution):100% EG, no acid added,100% EG, acidified with HCl (75 μL of 35–38% HCl per 150 g of EG),100% EG, acidified with acetic acid (750 μL of glacial acetic acid per 150 g of EG),50% EG, acidified with HCl (75 μL of 35–38% HCl per 150 g of 50% EG),50% EG, acidified with acetic acid (750 μL of glacial acetic acid per 150 g of 50% EG).

Each dispersion was aged at 40, 60, 80, and 100 °C for 1 month, and the variations in the ζ potentials and apparent particle size (measured at 25 °C) on aging were followed. This made a total of 20 parallel experiments: five different dispersions, each at four different temperatures. The dispersions aged at 40 °C were uniformly turbid and colorless during the entire experiment (1 month) except the pale yellow coloration that appeared in the samples with 100% EG, pH-neutral, and acidified with acetic acid. Similar pale yellow coloration appeared in all three dispersions in 100% EG aged at 60 °C, while two dispersions in 50% EG aged at 60 °C remained uniformly turbid and colorless during the entire experiment. The pale yellow color appeared after 1 day and the color did not change in time. Pale yellow coloration appeared in all five dispersions (50 and 100% EG) aged at 80 °C after one day, and in HCl-acidified dispersions (50 and 100% EG) the color became more intensive in the course of the experiment. In HCl-acidified dispersion in 100% EG, the final color of the dispersion was orange, and a brown deposit was precipitated. The pH-neutral and acetic-acid-acidified dispersions remained pale yellow and uniformly turbid during the entire experiment. Similar effects as at 80 °C were also observed at 100 °C, but the color of dispersions and of the deposit were more intensive (but still pale yellow), and the dark deposit was also observed in the acetic-acid-acidified dispersion in 100% EG. We are not sure how the pale yellow coloration may affect the potential performance of the dispersions as heat-transfer fluids.

The ζ potentials in dispersions aged at elevated temperatures are presented in Figure 9, Figure 10, Figure 11, Figure 12 and Figure 13.

Figure 9, Figure 10, Figure 11, Figure 12 and Figure 13 show that in all 20 experiments, the ζ potentials remained stable in time, and they were in excess of 40 mV (Smoluchowski); that is, in excess of 60 mV (Huckel). More consistent ζ potentials in 50% glycol and less consistent ζ potentials in 100% glycol resulted from high viscosity and low dielectric constant [22] in the latter system, as explained above for the dispersions aged at room temperature. Such high ζ potentials should be sufficient to protect the particles from coagulation and sedimentation, but visual observation led to the opposite conclusion in a few systems, as discussed above. The evolution of particle size in time in the same 20 dispersions is illustrated in Figure 14, Figure 15, Figure 16, Figure 17 and Figure 18.

Most values presented in Figure 14, Figure 15, Figure 16, Figure 17 and Figure 18 are in the range 60–100 nm; that is, they are similar to the apparent radii of primary particles presented in Figure 2 and Figure 5. The apparent radii shown in Figure 14, Figure 15, Figure 16, Figure 17 and Figure 18 are consistent with high absolute values of ζ potentials in these systems (Figure 9, Figure 10, Figure 11, Figure 12 and Figure 13). High ζ potentials, low particle radius, and the stability of these properties on aging at elevated temperatures suggest that the dispersions of amino-functionalized silica presented in Figure 9, Figure 10, Figure 11, Figure 12, Figure 13, Figure 14, Figure 15, Figure 16, Figure 17 and Figure 18 are suitable for heat-transfer fluids, and also for applications at temperatures up to 100 °C. Although only dispersions in 50 and 100% EG were studied at elevated temperatures, the present results suggest that acidified dispersions of amino-functionalized silica at any EG concentration between 50 and 100% also behave alike.

## 3. Materials and Methods

Fumed silica (Sigma Aldrich, Saint Louis, MO, USA) was 99.8% pure and its isoelectric point IEP, measured by means of different instruments, ranged from 2 to 4 [30]. The TEM (transmission electron microscopy) and FESEM (field emission scanning electron microscopy) images of the original fumed silica can be found elsewhere [31].

The 3-aminopropyltriethoxysilane (APTES, Thermo Fisher Scientific, Waltham, MA, USA) was 99% pure. The other chemicals were analytical grade from POCh, Lublin, Poland. The recipe from ref. [32] was used to functionalize silica. This recipe is similar to recipe 1 from our previous study [22], except larger amounts of reagents were used here.

Sartorius (Göttingen, Germany) PP-50 pH-meter with Oakton combination electrode (Cole Parmer, Vernon Hills, IL, USA) was used to measure the pH in aqueous systems. The combination electrode was calibrated with pH-buffers of pH 4, 7, and 9 from Chempur, Piekary Śląskie, Poland. The pH was not measured in EG or water–EG mixtures. The pH measurement and even the definition of pH in nonaqueous and mixed water–organic mixed solvents are difficult problems and they were discussed in more detail elsewhere [20].

Gemini V (Micromeritics, Norcross, GA, USA) was used to determine the specific surface area SSA (BET) of functionalized silica. The sample was dried at 100 °C before the SSA measurement to avoid possible degradation of surface amino groups at temperatures > 100 °C.

The amino-functionalized silica was characterized by IEP in aqueous dispersion. To this end, the particles were dispersed in NaCl solution at 25 °C (500 mg in 500 mL), and the electrophoretic mobility and particle size were measured with Malvern Zetasizer ZEN 3600 (Malvern, Malvern, England). Fresh and 3-day-aged dispersions adjusted to various pH levels were studied. Long-term behavior of three aqueous dispersions and of dispersions in pH-neutral and acidified aqueous glycol (30 mg in 100 g of solution) was studied using a method described in our previous study [22]. An ultrasonic bath was used to homogenize the freshly prepared dispersions. The dispersions were manually shaken just before collection of samples for the ζ potential and size measurements. This should emphasize that the ζ potential and particle size are rather insensitive to the concentration of solid particles [29]; thus, no special sampling procedure was needed. In this respect, ζ potential and particle size are very different from the measurements of viscosity and thermal conductivity (Table 1), where every small variation in the concentration of solid particles makes a big difference, and special sampling procedures are designed to assure a constant concentration of solid particles in a series of measurements. In specimens aged at elevated temperatures (40, 60, 80, and 100 °C), aliquots of dispersions (45 mg in 150 g of solution) were collected after specified time intervals, cooled to 25 °C, and the measurements were carried out at 25 °C, while the rest of the dispersion was further aged at an elevated temperature. The experiment was carried out without repetitions. Instead, we investigated many samples prepared under similar, but not identical, conditions, for example: EG concentrations every 10%, aging time every few days, pH every 0.5 pH unit, etc. We believe that similar properties (ζ potential, size) of dispersions, which differ slightly in their composition and age, confirm the reproducibility of our results equally well as compared to repetition of experiments under identical conditions. Moreover, the results produced by Malvern software (version 6.12) and reported here are in fact average values of multiple measurements (of the velocity of single particles). The instrument produces the ζ potential, along with a standard deviation, which is typically about 3 mV in pure water and about 30 mV in pure glycol, but the figures were already overcrowded, and we did not add error bars to the figures. We performed similar experiments with propylene glycol, and the results) were very similar as those with EG.

The ζ potential was calculated from electrophoretic mobility by means of the Smoluchowski equation, μ = εζ/η, where μ is the electrophoretic mobility, and ε and η are the dielectric constant and viscosity of the fluid. The dielectric constants and viscosities of aqueous glycol at 25 °C used to convert the mobility into ζ potential were taken from refs. [33,34], and their values are also reported in [22]. The Smoluchowski equation underestimates the ζ potential when the ionic strength is low and the particles are fine, which was the case in this study. Huckel equation, μ = 2/3 εζ/η, may be a better approximation in such systems. Huckel equation produces a ζ potential higher by a factor of 1.5 than Smoluchowski equation.

## 4. Conclusions

Although the present study was performed with one specimen of amino-functionalized silica, similar effects are probably common for other specimens. The present results show that dispersions of amino-functionalized silicas in EG and in aqueous EG show high positive ζ potentials (>40 mV), and the dispersions consist chiefly of primary particles (no aggregation). The positive ζ potential is especially high at EG concentrations > 40% and in acidified dispersions. These properties are preserved on aging up to 1 month at temperatures up to 100 °C. Application of acidified dispersions may have an adverse effect on the corrosion of metals, especially at high temperatures.

## Figures and Tables

**Figure 1 molecules-29-02686-f001:**
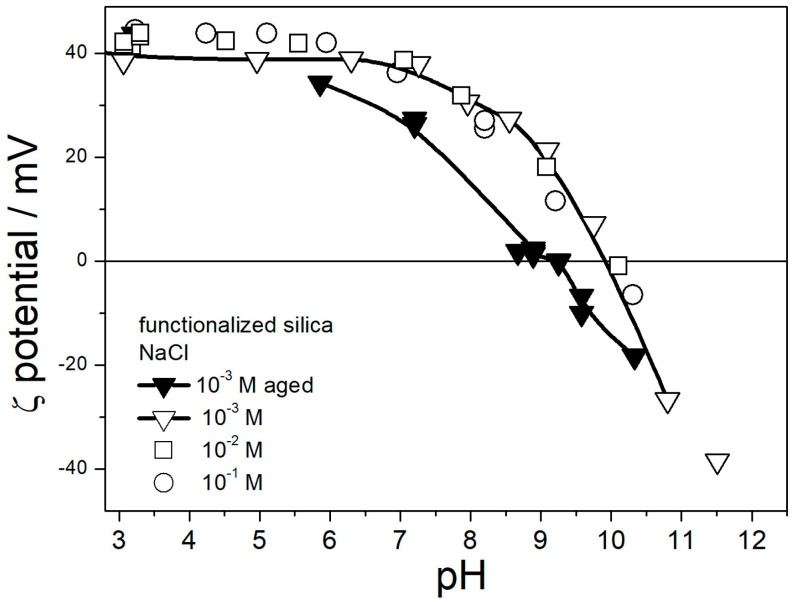
Zeta potential of amino-functionalized silica in aqueous 10^−3^–10^−1^ M NaCl. White symbols—fresh dispersions. Black symbols—3 d aged dispersions. The curves have been added to guide the eye.

**Figure 2 molecules-29-02686-f002:**
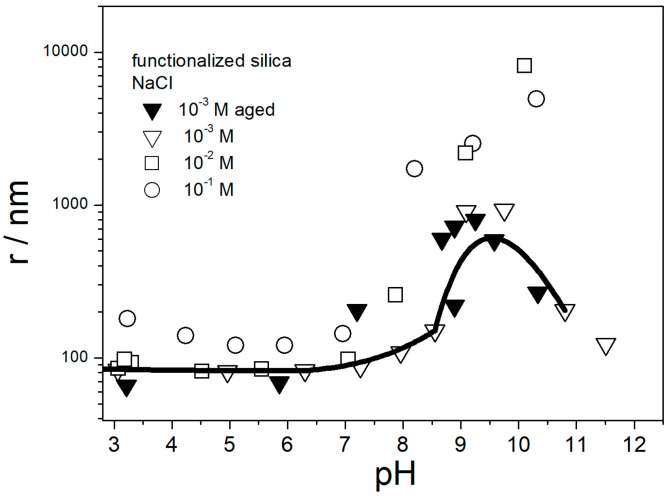
Apparent particle radius of amino-functionalized silica in aqueous 10^−3^–10^−1^ M NaCl. White symbols—fresh dispersions. Black symbols—3 d aged dispersions. The curve (10^−3^ M NaCl, fresh) has been added to guide the eye.

**Figure 3 molecules-29-02686-f003:**
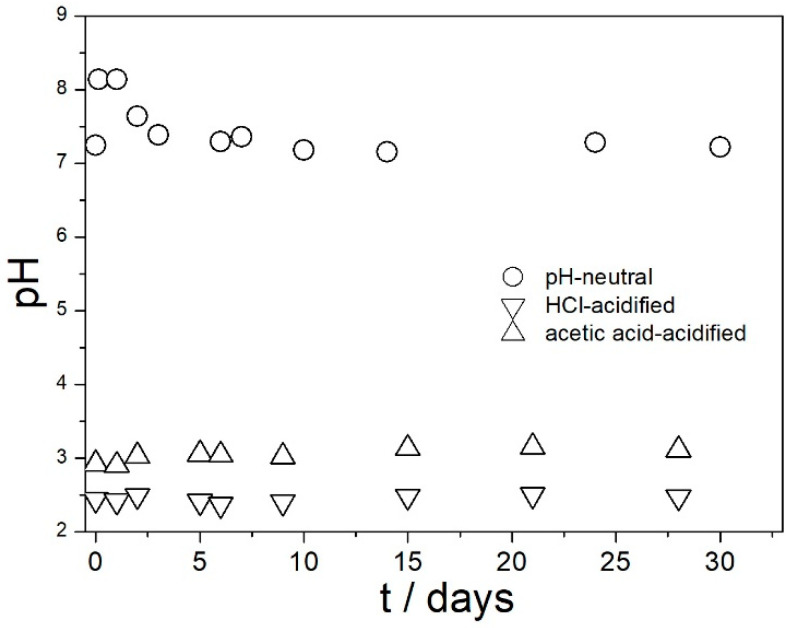
Evolution of pH in dispersions of amino-functionalized silica in water and in acid solutions.

**Figure 4 molecules-29-02686-f004:**
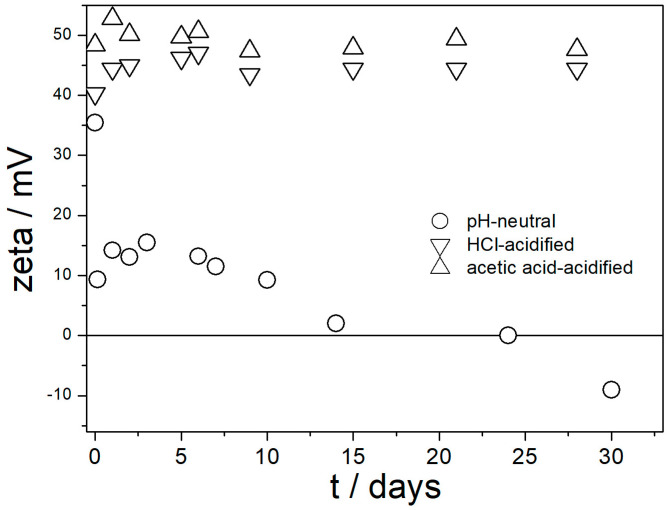
Evolution of zeta potential in dispersions of amino-functionalized silica in water and in acid solutions.

**Figure 5 molecules-29-02686-f005:**
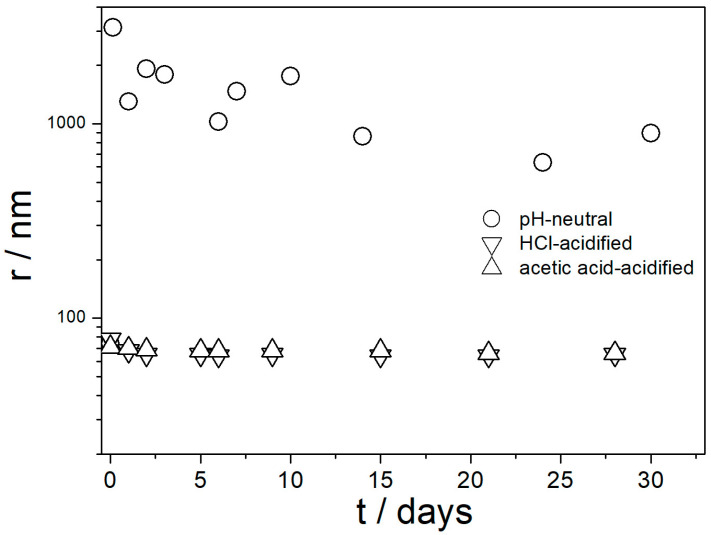
Evolution of apparent particle size in dispersions of amino-functionalized silica in water and in acid solutions.

**Figure 6 molecules-29-02686-f006:**
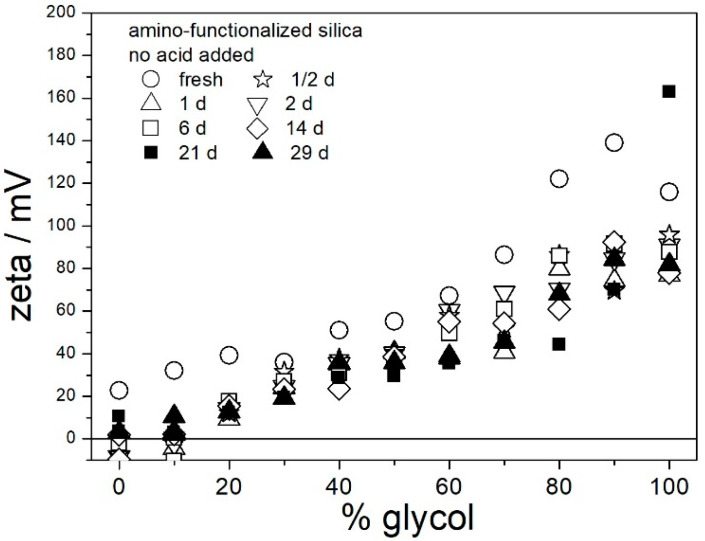
Evolution of ζ potential in dispersions of amino-functionalized silica (no acid added).

**Figure 7 molecules-29-02686-f007:**
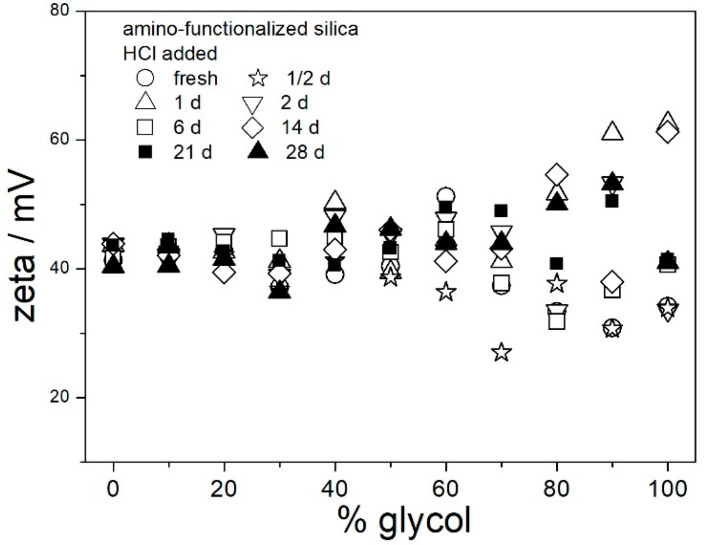
Evolution of ζ potential in dispersions of amino-functionalized silica in aqueous glycol acidified with HCl.

**Figure 8 molecules-29-02686-f008:**
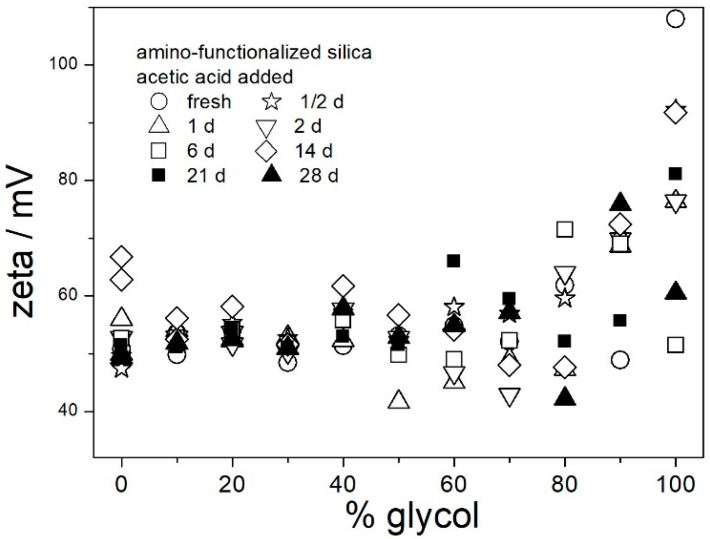
Evolution of ζ potential in dispersions of amino-functionalized silica in aqueous glycol acidified with acetic acid.

**Figure 9 molecules-29-02686-f009:**
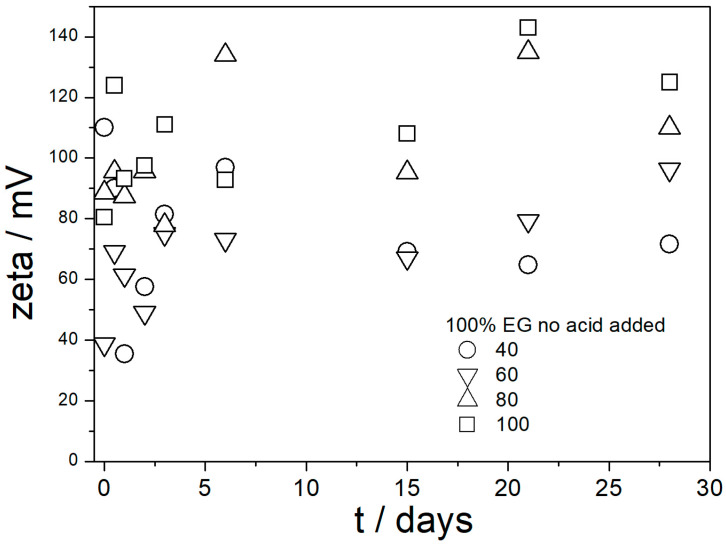
Evolution of ζ potential in dispersions of amino-functionalized silica in 100% EG (no acid added) at various temperatures (in °C).

**Figure 10 molecules-29-02686-f010:**
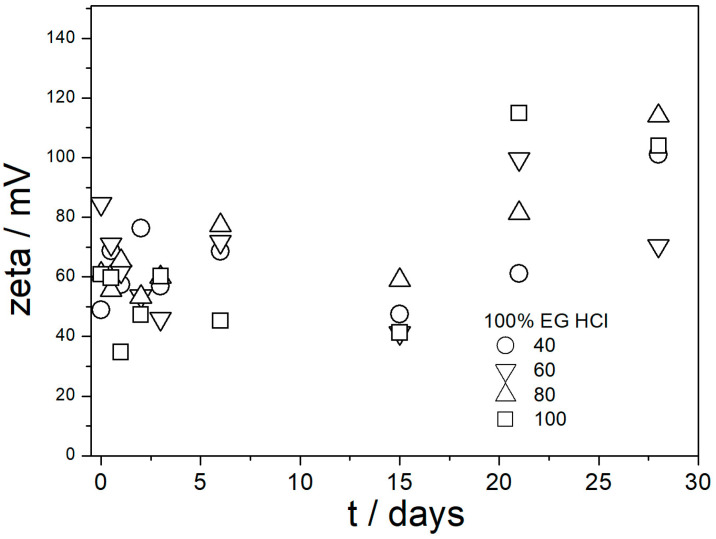
Evolution of ζ potential in dispersions of amino-functionalized silica in 100% EG acidified with HCl at various temperatures (in °C).

**Figure 11 molecules-29-02686-f011:**
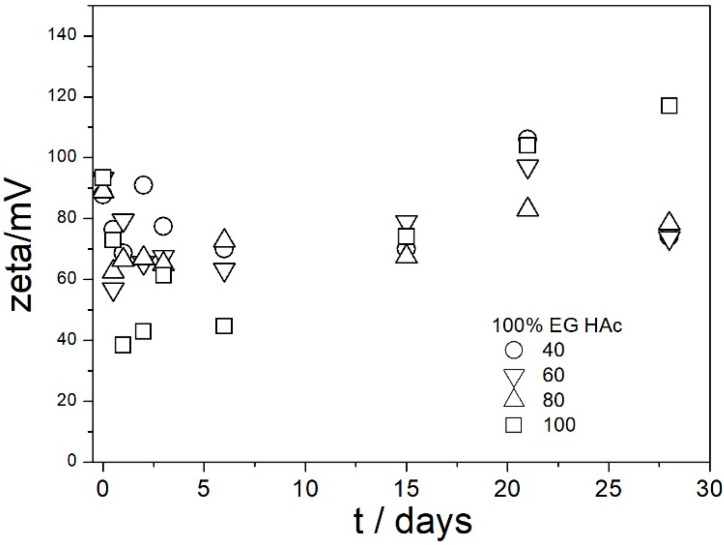
Evolution of ζ potential in dispersions of amino-functionalized silica in 100% EG acidified with acetic acid at various temperatures (in °C).

**Figure 12 molecules-29-02686-f012:**
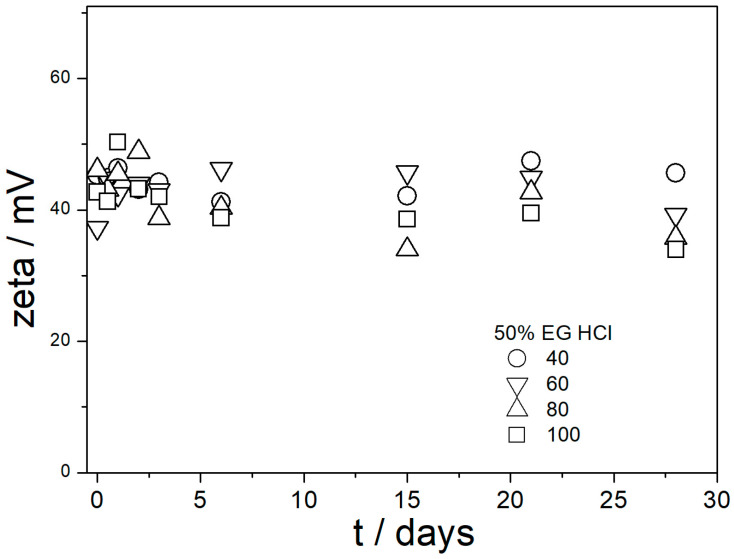
Evolution of ζ potential in dispersions of amino-functionalized silica in 50% EG acidified with HCl at various temperatures (in °C).

**Figure 13 molecules-29-02686-f013:**
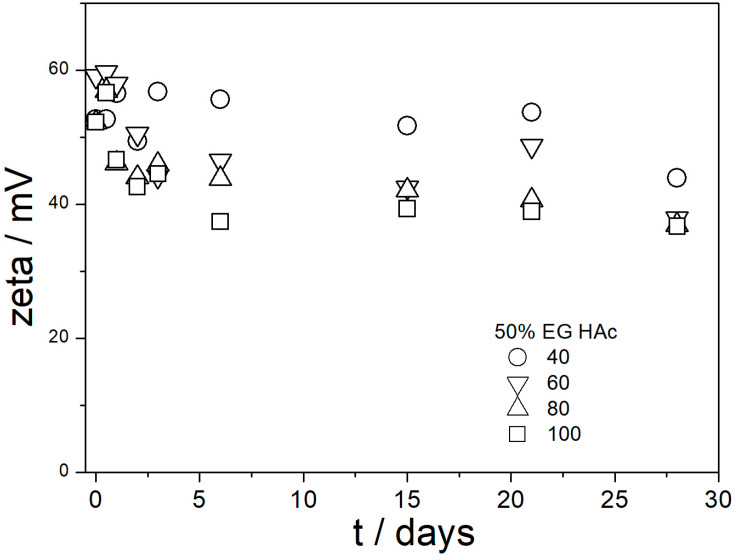
Evolution of ζ potential in dispersions of amino-functionalized silica in 50% EG acidified with acetic acid at various temperatures (in °C).

**Figure 14 molecules-29-02686-f014:**
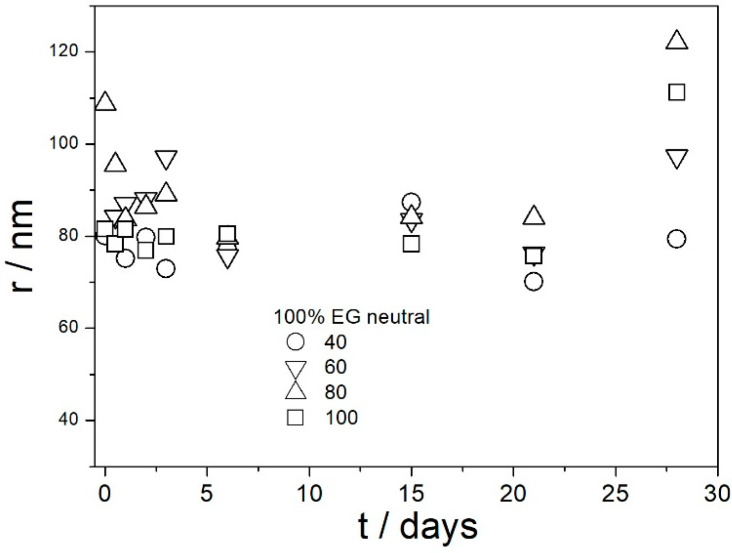
Evolution of apparent particle radius in dispersions of amino-functionalized silica in 100% EG (no acid added) at various temperatures (in °C).

**Figure 15 molecules-29-02686-f015:**
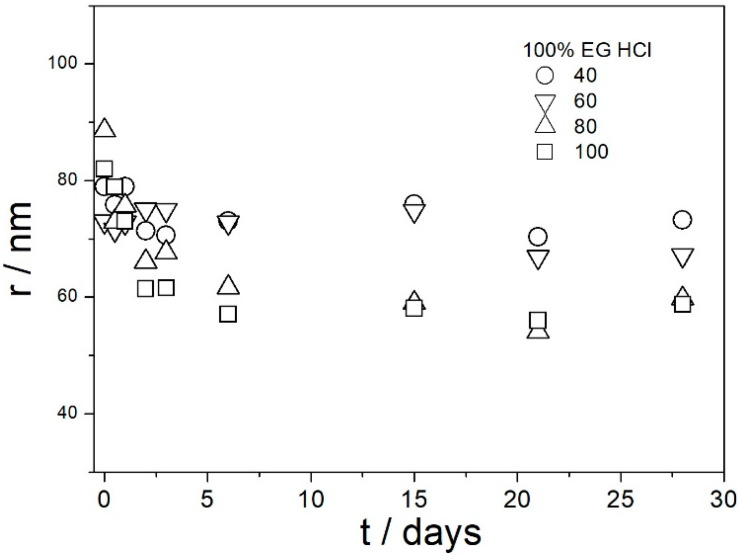
Evolution of apparent particle radius in dispersions of amino-functionalized silica in 100% EG acidified with HCl at various temperatures (in °C).

**Figure 16 molecules-29-02686-f016:**
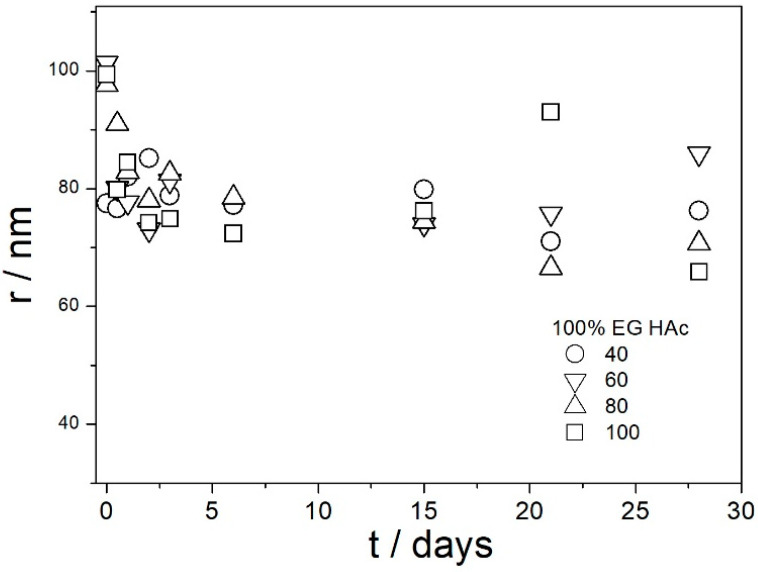
Evolution of apparent particle radius in dispersions of amino-functionalized silica in 100% EG acidified with acetic acid at various temperatures (in °C).

**Figure 17 molecules-29-02686-f017:**
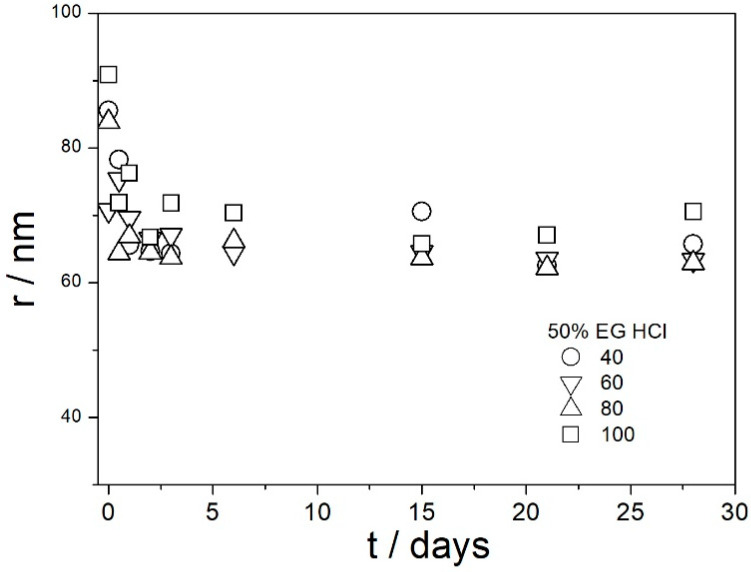
Evolution of apparent particle radius in dispersions of amino-functionalized silica in 50% EG acidified with HCl at various temperatures (in °C).

**Figure 18 molecules-29-02686-f018:**
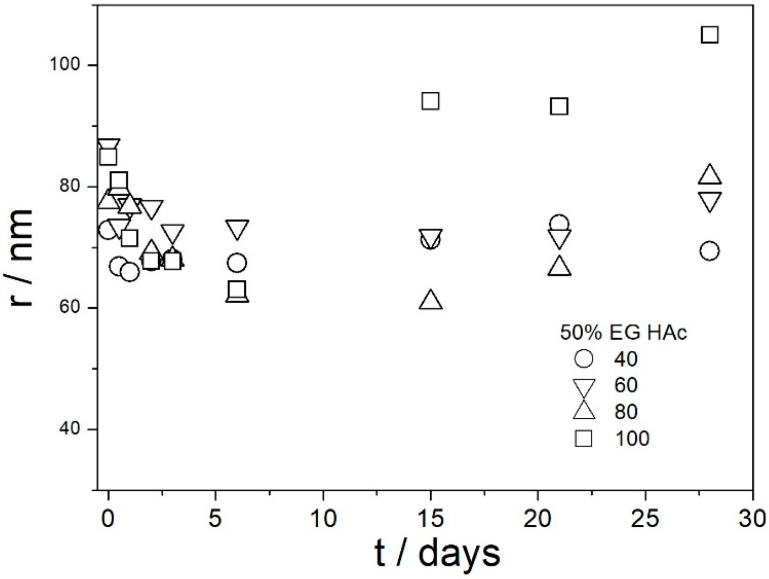
Evolution of apparent particle radius in dispersions of amino-functionalized silica in 50% EG acidified with acetic acid at various temperatures (in °C).

**Table 1 molecules-29-02686-t001:** Studies on viscosity and thermal conductivity of EG-based nanofluids. CNT = carbon nanotubes, GNP = graphene platelets.

Solvent	Particles	Stabilizer	Temperature Range/°C	Reference
EG	WS_2_	various and surfactant-free	25–70	[1]
40% EG	graphene-TiO_2_	various	20–70	[2]
40% EG	graphene	various	20–70	[2]
50%EG	ZnO-neem	none	20–50	[3]
50%EG	ZnO	none	20–50	[3]
40% EG	alumina-titania	none	20–50	[4]
40% EG	CNTs-TiO_2_	sodium dodecylbenzene sulfonate	25–70	[5]
50% EG	alumina-silica	none	20–60	[6]
50% EG	MgO	various	300–350 K	[7]
80–100% EG	multiwall CNTs	none	50–120	[8]
propylene glycol	multiwall CNTs	various	303–353 K	[9]
commercial coolant based on aqueous propylene glycol	alumina		30–60	[10]
80–100% EG	CNT-GNP original and oxidized	none	50–200	[11]
80–100% EG	CNT-GNP original and oxidized	various	35–170	[11]

## Data Availability

Data are available from the corresponding author on request.

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
