# Peer review of "Thermal Stability of Dispersions of Amino-Functionalized Silica in Glycol and in 50–50 Aqueous Glycol"

_molecules, 2024, doi:10.3390/molecules29112686_

Round 1
Reviewer 1 Report
Comments and Suggestions for Authors
Dear authors, the object of the study presented in this manuscript is of scientific and technological interest and could be a good contribution to the state of the art of nanofluids and, indeed, to the journal.
However, to be considered for publication, this manuscript must be improved both in terms of format and scientific rigor and must resolve certain serious aspects.
Below, I leave different comments and suggestions that I hope will help improve the manuscript. If these aspects are resolved, its publication can be evaluated.
After the review of the manuscript, I have some comments, doubts, and suggestions:
1. Introduction section: The introduction needs to be improved, especially the structure of the text, and a deeper literature review needs to be performed. Additionally, it can be more practical to put the information from lines 42 to 62 into a table, etc. On the other hand, the text is missing numerous references.
2. Results and discussions:
2.1. How many samples and sample repetitions have been performed? Nanofluids are inhomogeneous materials; it is well known that it is imperative to perform a good sampling to represent any value. If the provided results were performed only by one sample, the data and, consequently, the study is insufficient. If, on the other hand, the study has been carried out with several samples and repetitions, please add the sampling carried out in the methodology section and add the error bars to all graphs. Without this, the values are not credible.
2.2. It is difficult to interpret or see the tendencies in more of the figures. Please improve the figures or add tendency lines.
2.3. Have you corroborated the particle sizes with other complementary techniques? For example, characterization by SEM will improve the study substantially and corroborate the results.
2.4. In some parts of the manuscript, you refer to “high or low” temperature (or another property). Please revise the manuscript to avoid this kind of indeterminate adjectives and use more specific language.
3. Materials and methods:
3.1. All reagents used must be described.
3.2. The amount of sample and its preparation must be incorporated for each test.
Author Response
- Introduction section: The introduction needs to be improved, especially the structure of the text, and a deeper literature review needs to be performed. Additionally, it can be more practical to put the information from lines 42 to 62 into a table, etc. On the other hand, the text is missing numerous references.
The results were presented in a form of a table as suggested by the referee. We also added a few references to the recent work in the field.
- Results and discussions:
2.1. How many samples and sample repetitions have been performed? Nanofluids are inhomogeneous materials; it is well known that it is imperative to perform a good sampling to represent any value.
The following explanation was added:
Ultrasonic bath was used to homogenize the freshly prepared dispersions. The dispersions were manually shaken just before collection of samples for the ζ potential and size measurements. This should be emphasize that the ζ potential and particle size are rather insensitive to the concentration of solid particles [29] thus no special sampling procedure is needed. In this respect ζ potential and particle size are very different from the measurements of viscosity and thermal conductivity (Table 1), where every small variation in the concentration of solid particles makes a big difference, and special sampling procedures are designed to assure a constant concentration of solid particles in a series of measurements.
If the provided results were performed only by one sample, the data and, consequently, the study is insufficient. If, on the other hand, the study has been carried out with several samples and repetitions, please add the sampling carried out in the methodology section and add the error bars to all graphs. Without this, the values are not credible.
The following explanation was added regarding the repetitions.
The experiment was carried out without repetitions. Instead we investigated many samples prepared at similar, but not identical conditions, for example: EG concentrations every 10%, aging time every few days, pH every 0.5 pH unit etc. We believe that similar properties (ζ potential, size) of dispersions which differ slightly in their composition and age confirm the reproducibility of our results equally well as repetition of experiments at identical conditions. Moreover the results produced by Malvern software and reported here are in fact average values of multiple measurements (of the velocity of single particles). The instrument produces the ζ potential along with a standard deviation, which is typically about 3 mV in pure water and about 30 mV in pure glycol, but the figures are already overcrowded, and we do not add error bars to the figures. We performed similar experiments with propylene glycol, and the results (not reported here) were very similar as those with EG.
The results obtained in the present study (at room temperature) are further corroborated by their similarity with the results obtained in our previous study [22] with another type of amino-functionalized silica. Finally ζ potential (Smoluchowski) > 40 mV assures the stability of dispersion. The most important result of this study is that ζ potential does not fall below 40 mV on aging and the exact numerical value is less important.
We also added several details into the experimental section. We have not used any special sampling procedure, but it seems from our current and previous experience that sampling is not crucial with ζ potential and particle size. We agree that sampling may be crucial with studies of viscosity and thermal conductance, where the concentration of solid particles matters, but with ζ potential and particle size, small variations in the concentration of solid particles are less significant.
2.2. It is difficult to interpret or see the tendencies in more of the figures. Please improve the figures or add tendency lines.
We added the trend lines in Fig. 1 and 2. Regarding the other figures: they are overcrowded already and for example 8 extra lines in Fig. 6-9 will not help much.
2.3. Have you corroborated the particle sizes with other complementary techniques? For example, characterization by SEM will improve the study substantially and corroborate the results.
The following was added
The TEM (transmission electron microscopy) and FESEM (field emission scanning electron microscopy) images of the original fumed silica can be found elsewhere [31].
We do not have any left overs of the material so we cannot perform additional experiments with the same sample of functionalized silica.
2.4. In some parts of the manuscript, you refer to “high or low” temperature (or another property). Please revise the manuscript to avoid this kind of indeterminate adjectives and use more specific language.
Corrected (marked in red in the new Ms.)
- Materials and methods:
3.1. All reagents used must be described.
done
3.2. The amount of sample and its preparation must be incorporated for each test.
done
Reviewer 2 Report
Comments and Suggestions for Authors
The presented paper is interesting and can be searched for by scientists. However, before publication, some aspects should be improved:
1. There is a lack of material characterization (SEM/TEM nanoparticles (before functionalization and after).
2. The authors should provide a detailed description of sample preparation (including used equipment, duration of each step, etc.), which allows readers to repeat the procedure fully.
3. The measurement procedure should also be described in more detail, additionally, there is a lack of information about pH measurements.
4. All abbreviations should be explained with the first use (e.g. IEP, BET, etc.).
5. The authors refer to the Smoluchowski and Huckel equations, it would be good to present this equation in the manuscript.
Author Response
- There is a lack of material characterization (SEM/TEM nanoparticles (before functionalization and after).
The following was added
The TEM (transmission electron microscopy) and FESEM (field emission scanning electron microscopy) images of the original fumed silica can be found elsewhere [31].
We do not have any left overs of the material so we cannot perform additional experiments with the same sample of functionalized silica.
- The authors should provide a detailed description of sample preparation (including used equipment, duration of each step, etc.), which allows readers to repeat the procedure fully.
Many details were added to the new Experimental section (marked in red)
- The measurement procedure should also be described in more detail, additionally, there is a lack of information about pH measurements.
The following was added
Sartorius PP-50 pH-meter with Oakton combination electrode was used to measure the pH in aqueous systems. The combination electrode was calibrated with pH-buffers of pH 4, 7, and 9 from Chempur, Piekary Åšlaskie, Poland. The pH was not measured in EG or water-EG mixtures. The pH-measurement and even the definition of pH in nonaqueous and mixed water-organic mixed solvents are difficult problems and they were discussed in more detail elsewhere [20].
- All abbreviations should be explained with the first use (e.g. IEP, BET, etc.).
All abbreviations are explained now (changes marked in red)
- The authors refer to the Smoluchowski and Huckel equations, it would be good to present this equation in the manuscript.
The following was added:
Smoluchowski equation: μ=εζ/η where μ is the electrophoretic mobility, and ε and η are the dielectric constant and viscosity of the fluid. The dielectric constants and viscosities of aqueous glycol at 25 °C used to convert the mobility into ζ potential were taken from ref. [33,34], and their values are also reported in [22]. Smoluchowski equation underestimates the ζ potential when the ionic strength is low and the particles are fine, which is the case in this study. Huckel equation: μ=2/3 εζ/η may be a better approximation in such systems. Huckel equation produces a ζ potential higher by a factor 1.5 than Smoluchowski equation.
That is, both equations are reported explicitly.
Round 2
Reviewer 1 Report
Comments and Suggestions for Authors
Accept in present form
Reviewer 2 Report
Comments and Suggestions for Authors
The authors responded satisfactorily to my comments, I recommend accepting the article in its current form.